# Metallically gradated silicon nanowire and palladium nanoparticle composites as robust hydrogenation catalysts

Yoichi M. A. Yamada [1✉], Heeyoel Baek[1], Takuma Sato[1], Aiko Nakao[2] & Yasuhiro Uozumi [3]

Heterogeneous catalysis of alkenes to alkanes is of great importance in chemical industry, but more efficient and reusable heterogeneous catalysts are still demanded. Here, we report a metallically gradated composite of a silicon nanowire array and palladium nanoparticles which are reused for the hydrogenation of an alkene. The catalyst promotes the hydrogenation of stilbene with atmospheric hydrogen (0.1 MPa) to give diphenylethane quantitatively. The recovered catalyst can be reused, and mediates the reaction without loss of yield more than one hundred times, whereas the stability of Pd/C degrades rapidly over 10 cycles of reuse. The catalyst allows the hydrogenation of a variety of alkenes, including tetra-substituted olefins. Structural investigation reveals that palladium nanoparticles are metallically gradated onto the silicon nanowire array under mild conditions by agglomeration of palladium silicide, as confirmed by XAFS and XPS together with argon-ion sputtering. This means of metal agglomeration immobilization may be applicable to the preparation of a variety of metal nanoparticle catalysts.

[1] RIKEN Center for Sustainable Resource Science, Wako, Saitama 351-0198, Japan. [2] Bioengineering Laboratory, RIKEN, Wako, Saitama 351-0198, Japan. [3] Institute for Molecular Science (IMS), Okazaki, Aichi 444-8787, Japan. ✉email: ymayamada@riken.jp

f catalysts worked perpetually with high catalytic activity, the chemical industry and catalytic science would benefit. Toward achieving this ultimate goal of catalytic science, researchers have developed highly active and reusable heterogeneous catalysts. Hydrogenation is a very important reaction in chemical processes, and thus, the development of highly reusable catalysts for hydrogenation is highly important and challenging[1–4]. Heterogeneous catalysts have been prepared mainly in two ways: the first is the immobilization of the metal species on solid supports through surface adsorption. Pd/C and Pd on metal oxides ($SiO_2$, $Al_2O_3$, MgO, $CeO_2$, etc.) are typical examples of this type (Fig. 1a)[5–14]. The other is the immobilization of metal species in solid matrices of polymers, dendrimers, and organic frameworks via coordinative interactions, where the metal species are stabilized through the steric bulk of the frameworks and/or the coordinative interactions with heteroatoms (e.g. N, O, P, and S) (Fig. 1b)[15–25]. For example, we have developed convoluted polymeric metal catalysts that promote various cross-couplings and Huisgen cycloadditions at mol ppm level loadings[26–31]. However, the adsorption and coordination interactions in these catalysts are generally weaker than those resulting from covalent and metal bonds, and may thus cause metal leaching problems, particularly in industrial applications.

To address these challenges, we envisioned the immobilization of metal nanoparticles on supports via bonding with the alloy/agglomeration of the metal nanoparticles and supports. In such an approach, the metal nanoparticles would be strongly anchored to the support via metallic bonds (Fig. 1c), and should overcome the traditional issues and improve catalysis.

The surface structure of support materials is known to significantly affect the catalytic activity and selectivity of a transformation. Nanostructured materials are attractive candidates as supports for transition metal nanoparticles for preparation of catalysts for sustainable and efficient chemical processes[32]. We have been interested in nanoscale reactions on support materials for the development of catalytic reactions for efficient organic transformations. Recently, we reported the development of a silicon nanowire array and palladium nanoparticle hybrid (SiNA-Pd) for catalytic organic transformations[33,34]. Here we investigate the reusability of this system for the hydrogenation of olefins, where the catalyst can be readily reused more than one hundred times without loss of catalytic activity. In contrast, the catalytic activity of Pd/C for the same transformation decreases during its reuse, and afforded

much lower catalytic activity. Further investigation revealed that the palladium nanoparticles (PdNPs) were anchored to the silicon as palladium silicides via Pd–Si metallic bonds, and gradually form an agglomerated Pd/Si alloy layer as shown in Fig. 1c.

In this communication, we present the high reusability and catalytic activity of SiNA-Pd for the hydrogenation of olefins, including tetra-substituted alkenes. We also show evidence of the immobilization mode of the metal nanoparticles on the supports through gradated alloy formation/agglomeration of the metal nanoparticles and the supports.

## Results and discussion

**High reusability for hydrogenation**. SiNA-Pd was reproducibly prepared in accordance with our previous report (Fig. 2), where palladium nanoparticles were installed into SiNA by reduction of Pd(II) on the SiNA surface carrying terminal hydrogens (see Supplementary method for details)[35–41]. The hydrogenation of *trans*-stilbene (**1a**) was carried out with SiNA-Pd corresponding to 0.12 mol% of Pd under 0.1 MPa hydrogen gas conditions and afforded bibenzyl (**2a**) in quantitative yield. The recovered catalyst was reused without of loss of catalytic activity 150 times more in the same reaction and afforded **2a** quantitatively in all conversions (Fig. 3). Notably, the turnover number of the catalyst in the consecutive reactions reached 125000. No Pd was detected in the reaction mixture using ICP-MS analysis (detection limit: 0.021 ppb), which indicated the lack of Pd leaching during the catalytic transformation. In contrast, the catalytic activity of Pd/C decreased significantly during its consecutive reuse, which is in-line with the literature reports[42,43].

**Hydrogenation of various alkenes**. With a highly reusable catalyst in hand, the hydrogenation of various alkenes was evaluated with 500 mol ppm (0.05 mol%) Pd of SiNA-Pd (**1**/SiNA-Pd = 2000/1 (mol/mol)) (Fig. 4). In all of the reactions presented in Fig. 4, the catalyst was reused to show the generality of its reusability. When the reaction of the less reactive tetra-substituted alkenes **1b-c** was carried out (Supplementary Data 1), we were pleased to find that the reaction proceeded smoothly to give the corresponding alkane **2b–c** in >99% and 96% yields, respectively. The recovered catalyst promoted the reaction to give **2b** in >99% and 97% yields (2nd use of SiNA-Pd), and in >99% and 96% yields (3rd use of SiNA-Pd), respectively (Entries 1 and 2) (Supplementary Data 2). Tri- and di-substituted alkenes, including aliphatic olefins, as well as an alkyne and an imine (**1b–k**) were readily converted by the fresh and recovered SiNA-Pd under similar conditions to the corresponding products **2b–i** in 91–99% yields (Entries 3–10).

SiNA-Pd was applied to the hydrogenation of fatty acids and triolein under neat conditions (Fig. 5). Full conversion of unsaturated fatty acids to saturated fatty acids is important for preventing the formation of *trans*-fatty acids[44–49]. The hydrogenation of oleic acid (**1l**) was performed and gave stearic acid (**2j**), a fatty acid which activates high density lipoprotein (HDL) and decreases low density lipoprotein (LDL)[50], in >99% yield (Fig. 5a). SiNA-Pd was reused under similar conditions and gave **2j** in >99% yield. Linoleic acid (**1m**) was also converted to **2j** in >99% yield (Fig. 5b). Triolein (**1n**) also underwent hydrogenation to give glyceryl tristearate (**2k**) in 99% yield (Fig. 5c).

In SiNA-Pd, the palladium nanoparticles are immobilized by the reduction with the silanes on the silicon surface. We hypothesized that the palladium nanoparticles were metallically gradated onto the silicon nanowire array via strong metal bonding (alloy/ agglomeration) between the palladium and silicon. Although we attempted studying the metal bonding in SiNA-Pd to gain insight into the immobilization mode of the

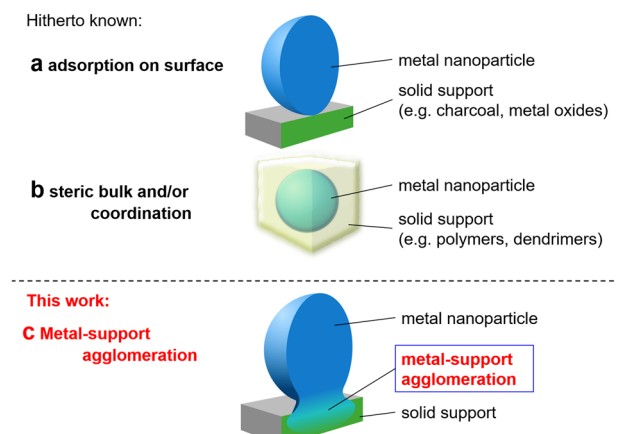

**Fig. 1 Morphology of metal-support stabilization in solid-supported metal nanoparticle catalysts. a** Surface adsorption. **b** Steric bulk and/or coordination. **c** Gradient agglomeration of metal nanoparticle and support material (this work).

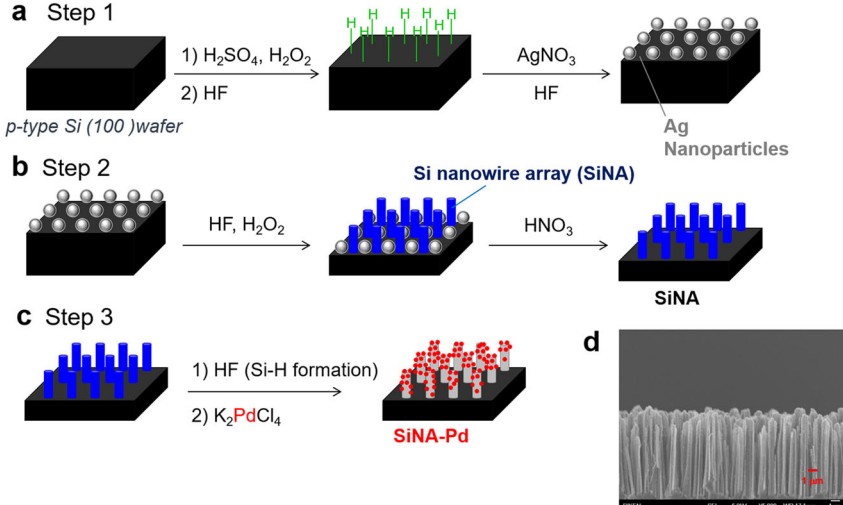

**Fig. 2 Schematic illustration for preparation of SiNA-Pd. a** Hydrogen-termination of Si(100) surface, followed by deposition of silver nanoparticles. **b** Formation of SiNA by metal-assisted chemical etching. **c** Preparation of SiNA-Pd through reductive deposition of palladium species. **d** SEM image of SiNA-Pd.

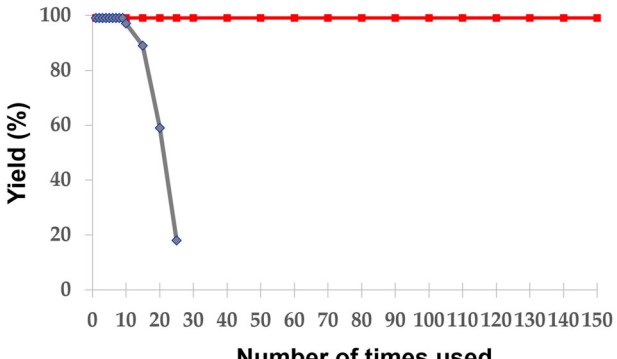

**Fig. 3 Hydrogenation of trans-stilbene (1a) for comparison with catalytic activity of SiNA-Pd and Pd/C.** *trans*-stilbene (**1a**) (0.5mmol), catalyst (0.12mol%), ethanol (2.0mL) under H$_2$ (0.1MPa) at 70°C for 24h. Red: SiNA-Pd; blue: Pd/C.

metal nanoparticles on supports, the spectroscopic analysis was not feasible due to the high steric crowding of SiNA-Pd with palladium nanoparticles and silicon nanowires. To address this limitation, we prepared less crowded silicon-nanostructured palladium nanoparticles under similar conditions for the following investigation.

**Elucidation of stable Pd–Si nanoparticle formation in the catalyst.** X-ray photoelectron spectroscopy (XPS) allows relatively high surface-sensitive spectroscopic analysis, because the mean free path of photoelectrons in solid materials is within several nanometers from the outermost surface, and the technique is often utilized in combination with sputtering techniques for the purpose of depth analysis[51]. Remarkably, a series of XPS spectra of the silicon-nanostructured palladium nanoparticles for Pd 3d core level exhibited the presence of palladium silicide[52] besides Pd(0) (Fig. 6). Ar-ion-sputtering was performed with 3-keV kinetic energy in vacuum. The XPS spectra were recorded at a take-off angle of 90˚. The as-prepared silicon-nanostructured palladium nanoparticles were etched by sputtering for up to 16 min in vacuum, and the nominal etching rate was determined to be 0.5 nm/min of SiO$_2$

using a SiO$_2$/Si standard. With the increase in the sputtering time, the peak intensity of Pd(0) (ca. 335 eV for Pd 3d$_{5/2}$) decreased, whereas the intensity in the higher binding energy region (similarly 336–338 eV) increased. The surface oxides on PdNPs (usually described as Pd(II)O and Pd(IV)O$_2$ (It is known that palladium dioxide synthesized at a high oxygen pressure decomposes at temperatures above 340 K)[53,54]) should be removed from the surface of the sample after the etching. Indeed, a weak and very broad peak with a significantly high binding energy (>338 eV for Pd 3d$_{5/2}$) almost completely disappeared after a sputtering time of 1 min, which is attributed to the surface PdO$_2$ localized within the block boundaries of PdO[55]. We suppose that the reason for the increased intensity in the high binding energy region in the 335–338 eV range for Pd 3d$_{5/2}$ is due to the depth distribution of the palladium species. Thus, the higher binding energy species localize behind the Pd(0) nanoparticles. Furthermore, the binding energy of the main peak near 336 eV in the spectra of the etched samples was not consistent with that of PdO in the surface-oxidized palladium foil (336.9 eV)[55]. Over a period of the sputtering, the intensity of the broad peak derived from SiO$_2$ near 102.6 eV in the spectra for Si 2p core level decreased monotonically relative to that of bulk silicon, which indicates the etching of the SiO$_2$ surface along with that of the palladium oxide and bulk palladium. In addition, the peak shape of the crystalline silicon (split into 2p$_{1/2}$ and 2p$_{3/2}$ due to spin–orbit coupling) in the 98–100 eV range was continuously deformed during the etching. As is the case with a series of Pd 3d spectra, this indicated that the contribution of palladium silicide increased in the early stage of the etching, which affected the peak shape around 99 eV in the Si 2p spectra due to the removal of Pd(0).

We were successful in separating the peak components of Pd (0), and palladium silicide (Pd$_4$Si, Pd$_3$Si, Pd$_2$Si, and PdSi) in the Pd 3d spectrum at a sputtering time of 4 min (Fig. 7). The envelope was fitted at 334.6/339.8 eV (3d$_{5/2}$/3d$_{3/2}$) for Pd(0), 335.4/340.6 eV for Pd$_4$Si, 336.1/341.3 eV for Pd$_3$Si, 336.9/342.1 eV for Pd$_2$Si, and 337.8/343.08 eV for PdSi. The values of full width at half maximum (FWHM) were within the range of 0.9–1.3 eV, and these values of binding energy and FWHM were consistent with those reported in earlier research[55]. According to the observations in Fig. 6, in which the population of Pd 3d$_{5/2}$ peaks gradually shifted toward the higher binding energy region with an

| Entry | Substrate | Product | Yield (%) (1st use of SiNA-Pd) | Yield (%) (2nd use of SiNA-Pd) |
|---|---|---|---|---|
| 1[b] | 1b | 2b | >99 | >99 (>99)[c] |
| 2[b] | 1c | 2c | 96 | 97 (96)[c] |
| 3 | 1d | 2d | 99 | 98 |
| 4 | 1e | 2a | 98 | 96 |
| 5 | 1f | 2e | 99 | 99 |
| 6 | 1g | 2f | 99 | 99 |
| 7 | 1h | 2a | 96 | 91 (97)[c] |
| 8 | 1i | 2g | 97 | 97 (97)[c] |
| 9 | 1j | 2h | 91 | 91 |
| 10 | 1k | 2i | >99 | >99 |

**Fig. 4 Hydrogenation of alkenes by using fresh and used SiNA-Pd under hydrogen atmosphere (0.1 MPa). a** Conditions: 1 (0.5 mmol), $H_2$ (0.1 MPa), SiNA-Pd (500 mol ppm Pd), EtOH (2 mL), 70 °C, 12 h. **b** The reaction time: 24 h. **c** Yield in the 3rd use of SiNA-Pd.

increase in the etching time, the more silicon-rich silicide was distributed far from the outermost surface than the palladium-rich silicide and Pd(0). We also performed curve fitting on the Si 2p spectrum at the sputtering for 4 min to separate the peak components of crystalline silicon ($2p_{1/2}$ and $2p_{3/2}$), palladium silicide, $SiO_x$ (suboxide), and $SiO_2$. The envelope was fitted at 98.4 eV for $2p_{3/2}$, 99.2 eV for $2p_{1/2}$, 99.3 eV for $Pd_xSi$, 101.0 eV for $SiO_x$, and 102.6 eV for $SiO_2$. The values of FWHM are 0.7, 0.7, 2.3, 1.7, and 1.7 eV, respectively. Due to the very close chemical shifts of palladium silicide[56–58], we let a somewhat broad peak (FWHM is 2.3) represent the sum of palladium silicides. The peak deformation near the crystalline silicon during the etching is possibly attributed to the increasing contribution of the palladium silicide due to the removal of Pd(0) from the surface toward the silicide phase.

We evaluated the chemical state of the palladium in silicon-nanostructured palladium nanoparticles by Pd $L_3$-edge X-ray absorption near-edge structure (XANES), which is often used to probe the unfilled $d$ orbitals of transition metals (Fig. 8)[59–61]. (The surface of Pd foil was cleaned by Ar-ion-sputtering prior to the measurement to remove the surface oxide layer of palladium. The samples were transported for analysis at BL-13 beamline of Ritsumeikan SR Center (Shiga, Japan) without air-exposure.) The curve fitting for the near-edge region in the spectrum of silicon-nanostructured palladium nanoparticles was also performed to distinguish the chemical species in the higher-energy region to Pd (0) in the Pd $L_3$-edge spectrum. Two pseudo-Voigt functions and one error function were applied to the two peaks (one of them is Pd (0)) and baseline (edge-jump of Pd $L_3$-edge), respectively. The best fit with two peaks ($E_0 = 3174.2$ and 3177.6 eV, where $E_0$ is the

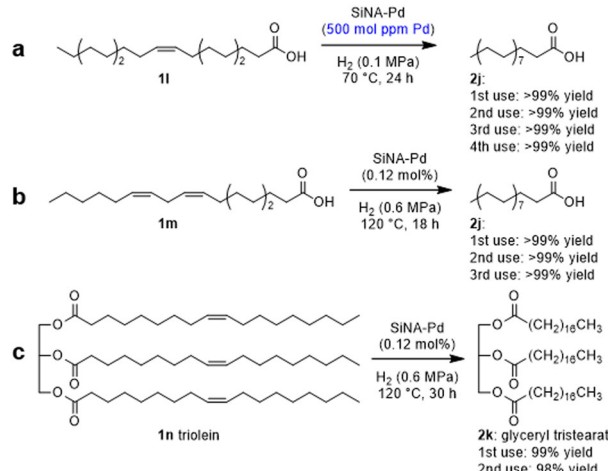

**Fig. 5 Hydrogenation of fatty acids. a** Oleic acid (0.5 mmol), SiNA-Pd (500 mol ppm), $H_2$ (0.1 MPa), 70 °C, 24 h. **b** Linoleic acid (0.5 mmol), SiNA-Pd (0.12 mol%) $H_2$ (0.6 MPa), 120 °C, 18 h. **c** Triolein (0.5 mmol), $H_2$ (0.6 MPa), 120 °C, 30 h.

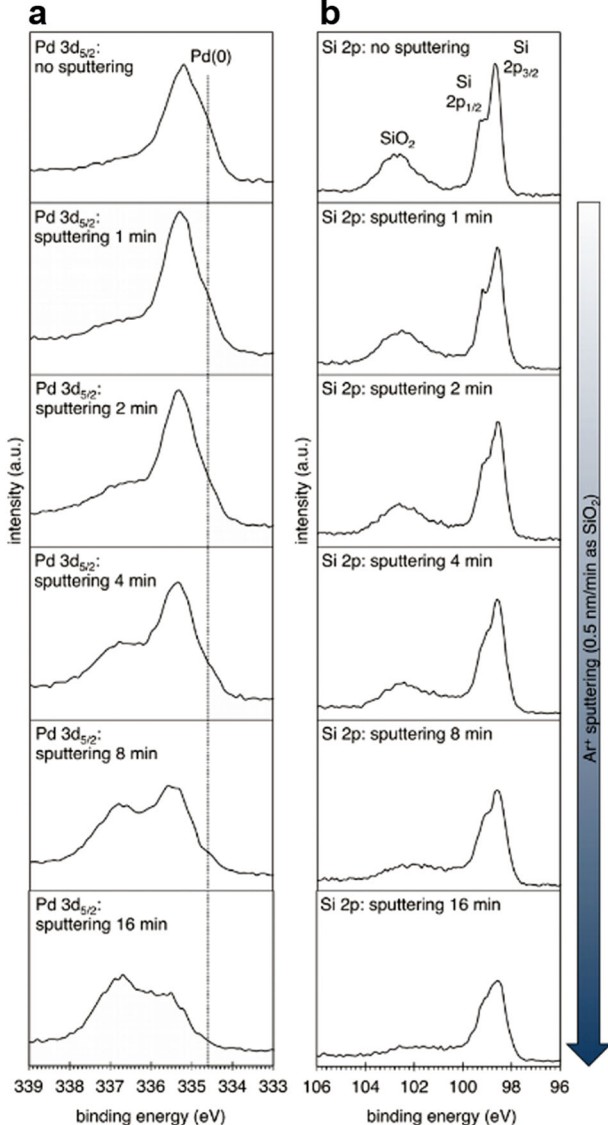

**Fig. 6 Study of Pd $3d_{5/2}$ and Si 2p XPS spectra.** Pd $3d_{5/2}$ (**a**) and Si 2p (**b**) XPS spectra of the silicon-nanostructured palladium nanoparticles at a take-off angle of 90° with Ar-ion-sputtering with 3-keV kinetic energy. The nominal etching rate was 0.5nm/min as $SiO_2$. The binding energy of Pd(0) in Pd $3d_{5/2}$ core level is displayed as dotted line at 334.6eV based on the curve fitting result for the spectrum at the sputtering time of 4min (Fig. 7).

centroid energy of line) showed good agreement (R-factor: 0.4%) with the experimental line shape of the silicon-nanostructured palladium nanoparticles in the range 3165–3180 eV. The $E_0$ value of 3174.2 eV (fixed during fitting) is identical to that obtained from the fitting result for the spectrum of Pd foil. As expected, another peak (optimized for all parameters) appeared at a significantly higher $E_0$ (3177.6 eV) and was slightly broader compared with Pd(0); this peak was assigned to the sum of palladium silicides. Additionally, the IR experiments indicated that after the formation of palladium nanoparticles on the HF-treated silicon surface, the surface Si–H were completely consumed and an oxidized surface of silicon was generated (Supplementary Fig. 1).

Taking all these observations into account, we proposed an immobilization mechanism for the reduction of Pd(II) species onto the silicon surface comprising terminal hydrogens (Fig. 9). The initial step in the formation of PdNPs is the generation of palladium silicides, which are formed in the reaction of Pd(II) with the surface Si–H$_x$. Subsequently, more palladium-rich silicide species (Pd$_2$Si, Pd$_3$Si, and Pd$_4$Si) are also generated, and the PdNPs grow on the silicides associated with the reduction of Pd(II) by the surface Si–H and Si–Si bonds. The latter pathway (reduction of Pd(II) by the cleavage of Si–Si bond) can explain the reason for the mismatch between the amounts of surface hydrides and PdNPs. Thus, Pd(II) species are reduced by not only the surface Si–H but also Si–Si bonds near the surface. The silicon substrate serves as a reducing agent and the electrons are provided through the palladium silicide layer to the outermost surface of PdNPs at which the reduction of Pd(II) to Pd(0) occurs. Overall, the immobilization of palladium onto silicon is so robust and stable that the catalyst is reusable more than one hundred times without loss of catalytic activity.

We found that our silicon nanowire array-stabilized palladium nanoparticle catalyst (SiNA-Pd) can be reused more than one hundred times without loss of catalytic activity in alkene hydrogenation, and affords the corresponding alkanes quantitatively. A variety of alkenes including tetra-substituted olefins were readily converted to alkanes in high yield by fresh and reused SiNA-Pd. Metallically gradated Pd–Si nanoparticles formed on SiNA-Pd provided perpetual heterogeneous catalysis of hydrogenation. Since this methodology for immobilizing metal nanoparticles onto silicon nanowire arrays is promising, we are developing other metal catalysts for organic transformations by utilizing our methodology.

## Methods

**General information.** See Supplementary Methods.

**General procedures.** See Supplementary Methods.

**Characterization.** See Supplementary Figs. 2–26.

**X-ray structure determination.** See Supplementary Fig. 29 and Supplementary Fig. 30. CIFs are available in Supplementary Data 1–2.

**IR Experiments.** See Supplementary Fig. 1.

**Preparation of a SiNA-stabilized Pd nanoparticle catalyst.** A boron doped p-type Si (100) wafer (0.1–100 Ω cm; Φ = 2 inch; 1.5 g; $S_{BET}$ 4 cm$^2$/cm$^2$ (Kr)) was immersed in a mixture of conc. 95% $H_2SO_4$ and 30% $H_2O_2$ (15 mL: 5 mL, v/v) for 15 min, and then, washed with $H_2O$ and dried with $N_2$ blow. The washed Si wafer was treated with 5% aqueous HF (10 mL) for 3 min, washed with $H_2O$, dried with

N$_2$ blow. One side of Si wafer was masked with a urethane mask sheet (Kokuyo, Co. Ltd). Masked Si wafer was placed into a mixture of 46% HF (15 mL) and AgNO$_3$ (53.2 mg) in H$_2$O (47.6 mL), which was slowly stirred for 1 min. The Ag-coated Si wafer was washed with H$_2$O, and dried with N$_2$ blow. The urethane mask sheet was peeled off. The Ag-coated Si wafer was placed into a mixture of 46%HF (4.0 mL) and 30% H$_2$O$_2$ (0.9 mL) in H$_2$O (19.0 mL) at 60 °C for 3 min, and Si wafer was washed with H$_2$O and dried with N$_2$ blow. The etched Si wafer was immersed

twice in 50% aqueous HNO$_3$ (30 mL) for 3 min, washed with H$_2$O, dried with N$_2$ blow to give the Si nanowire array (SiNA). After SiNA was placed into 5% aqueous HF (10 mL) for 1 min, washed with H$_2$O, dried with N$_2$ blow, it was immersed in a mixture of 50 mM aqueous K$_2$PdCl$_4$ (4.5 mL) and acetone (1.5 mL) for 5 min. The Si wafer was washed with H$_2$O and acetone, and dried with N$_2$ blow to give SiNA-stabilized Pd nanoparticle catalyst (SiNA-Pd). The loading of Pd was determined by ICP-MS.

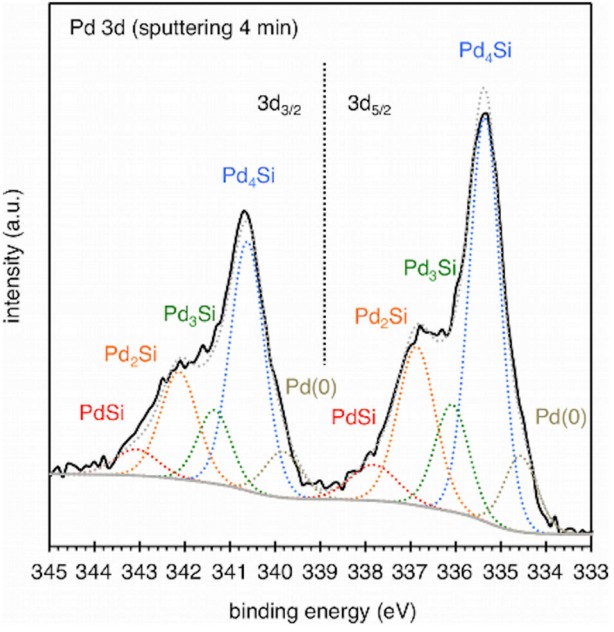

**Fig. 7 Pd 3d XPS spectrum of silicon-nanostructured palladium nanoparticles.** The black solid line shows the result of the sputtering time of 4 min, and the colored dotted lines do the curve fitting results for Pd(0), Pd$_4$Si, Pd$_3$Si, Pd$_2$Si, and PdSi.

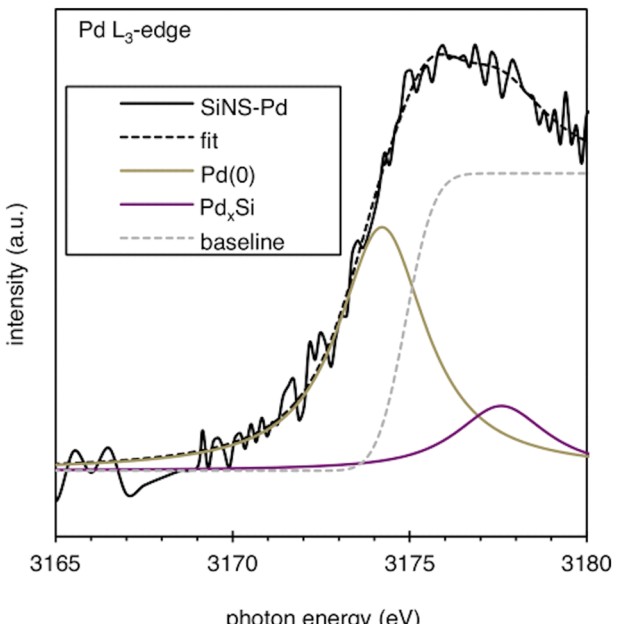

**Fig. 8 Curve fitting for the white line in the Pd L3-edge XANES spectrum of silicon-nanostructured palladium nanoparticles.** Two pseudo-Voigt functions are applied to Pd(0) and Pd$_x$Si.

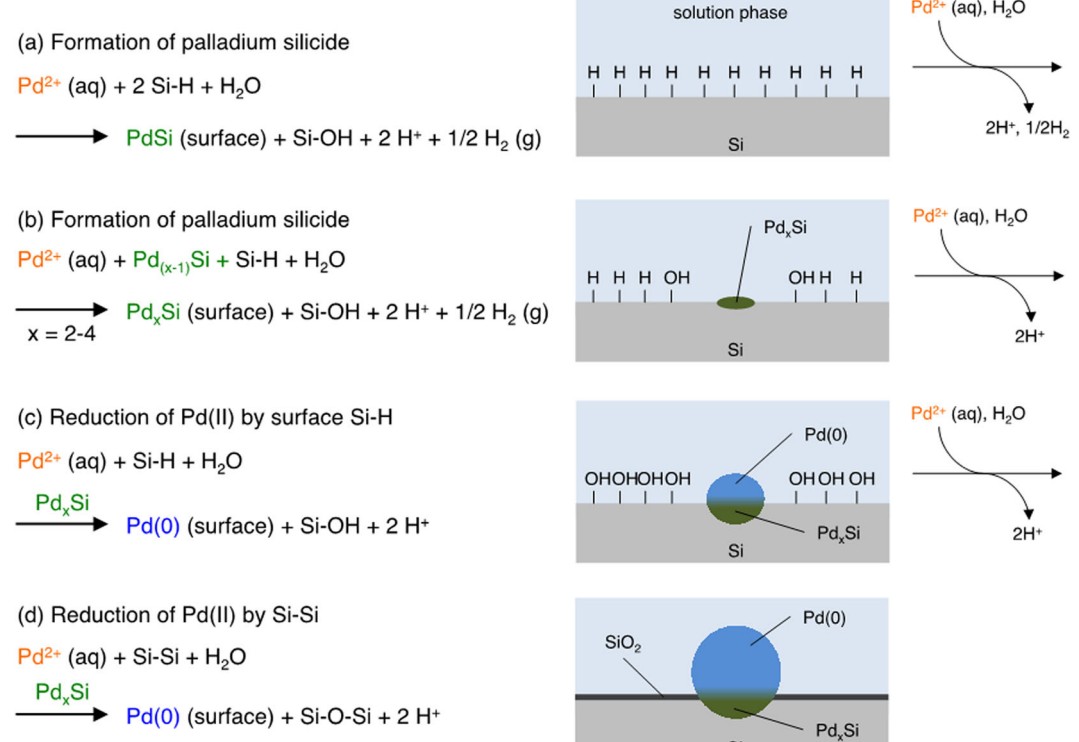

**Fig. 9 Plausible mechanism and illustration for immobilization of Pd nanoparticles on silicon surface bridged by palladium silicide layer. a** Initial formation of palladium silicide. **b** Consecutive formation of palladium silicide. **c** Reduction of Pd(II) by surface Si–H. **d** Reduction of Pd(II) by Si–Si.

**General procedure for the hydrogenation of various substrates**. To a 10 mL glass vessel was added SiNA-Pd (0.00025 mmol, 0.05 mol% Pd), each substrate (0.5 mmol), and ethanol (2 mL). The reaction was carried out with $H_2$ (0.1 MPa, balloon) at 70 °C for 12–24 h under the vortex mixing conditions by a vortex mixer with a temperature controller. After cooling to room temperature, the reaction mixture was analyzed with GC (HP-1) with dodecane or mesitylene as an internal standard to determine the yield.

**Catalyst reuse experiments**. To a 10 mL glass vessel was added SiNA-Pd (0.00025 mmol, 0.05 mol% Pd), *trans*-stilbene (0.5 mmol, 90 mg), ethanol (2 mL), and dodecane as an internal standard. The reaction was carried out with $H_2$ (0.1 MPa, balloon) at 70 °C for 24 h under vortex mixer. After cooling to room temperature, the reaction mixture was analyzed with GC (HP-1) to determine the yield. The catalyst was recovered by picking up with a tweezer and washed with EtOAc/acetone/water. After drying with $N_2$ blow, the catalyst was used for the next reaction.

## Data availability

All data supporting the findings of this study are available within the paper as well as the Supplementary Information file, or available from the corresponding authors on reasonable request. The X-ray crystallographic coordinates for structures reported in this Article have been deposited at the Cambridge Crystallographic Data Centre (CCDC), under deposition number CCDC1939498 and CCDC1939526. These data can be obtained free of charge from The Cambridge Crystallographic Data Centre via www.ccdc.cam.ac.uk/data_request/cif.

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

## Acknowledgements

The authors would like to thank Dr. Daisuke Hashizume (Materials Characterization Support Team, CEMS, RIKEN, Japan) for X-ray crystallography, Dr. Tetsuo Honma (JASRI) for hard X-ray XAFS experiment, and Ms. Aya Ohno (our team) for ICP-MS analysis. We are also very grateful to Prof. Toshiaki Ohta and Dr. Keisuke Yamanaka (SR Center, Ritsumeikan University) for soft X-ray XAFS experiments. Mass spectral data were acquired at the mass spectrometry facility run by Molecular Structure Characterization Unit (RIKEN CSRS, Wako). We gratefully acknowledge financial support from the JST ACT-C (#JPMJCR12ZC), the JST ACCEL (#JPMJAC1401), the JSPS (#24550126, #20655035, and #15K05510), AMED (#19ak0101115h), the Takeda Science Foundation, the Naito Foundation, and RIKEN.

## Author contributions

Y.M.A.Y. and Y.U. conceived the work and designed the experiments. H.B. carried out the catalytic reactions. A.N. conducted the XPS measurements. T.S. prepared the silicon Pd devices for XPS measurements and the spectroscopic measurements. All the authors contributed to intellectual insights of this work. Y.M.A.Y., H.B., and T.S. wrote the paper, and all authors edited the paper.

## Competing interests

The authors declare no competing interests.
