## [Peer Review File · Communications Chemistry]

Reviewers' comments:

Reviewer #1 (Remarks to the Author):

This paper introduces significant improvements in hydrogenations procedure that can be of interest to others in the community and the wider field. Results and conclusion are convincing. In my opinion, the paper deserves to be published on this Journal after some minor revision/request listed into the attached review file.

Reviewer #2 (Remarks to the Author):

Yamada and coworkers report a methodology of immobilization of a palladium catalyst on to a silicon nanowire array and its usage in the alkene hydrogenation including tetrasubstituted substrates. The immobilized Pd catalyst is highly active for the hydrogenation of alkenes giving the corresponding saturated products in high yields. They also show an application of hydrogenation of unsaturated fatty acids to form the saturated ones without the contamination by trans-fatty acids. The authors also show the reusability of the catalyst by recycling the catalyst up to 150 times and almost no leaching of the Pd metal by ICP-MS analysis. They propose the formation of Pd-Si alloy (palladium silicide) on the immobilization of the catalyst based on the XPS analysis and XANES spectra of Pd on silicon nanowire array which has low density of Pd on SiNA as a model. Although the authors have previously reported the immobilization method of a metal catalyst on SiNA and its reusability, this article includes some new findings which are worth publishing in Communications Chemistry. Concerning the terms "amalgam" and "amalgamation", these terms are obscure because these are often used only for the formation of mercury alloy.

Reviewer #3 (Remarks to the Author):

This paper reported a perpetually reusable heterogeneous catalyst for alkene hydrogenation through metal amalgamation immobilization of Pd nanoparticles on silicon nanowire arrays. Though the heterogeneous catalyst of SiNA-Pd have been used for catalytic organic transformations in the author's previous works (ref. 33 in this paper), the author found this heterogeneous catalyst could be reused without yield-loss for more than one hundred times and investigated the mechanism for the stability in this paper. This metal amalgamation immobilization may be applied to the preparation of a variety of metal nanoparticle catalysts with high stability. The results are interesting. I think this paper should be published after addressing the following comments:

1. The scheme of fabricating processes as shown in fig.2a is the same as scheme 1 in ref. 33.
2. The characterization of the heterogeneous catalyst of SiNA-Pd is inadequate. Both magnified SEM and TEM images are needed to intuitively show the structures of the heterogeneous catalyst. Especially, high-resolution TEM may show the lattice transition from silicon to Pd particles.
3. The authors showed compared results of Pd/C catalysts in Fig. 3, but didn't show the structure of the Pd/C catalyst or the related description of the contrast experiment. In addition, why does the yield of Pd/C catalyst sharply decrease after 10 cycles?
4. Could the heterogeneous catalyst be formed through metal amalgamation immobilization by using a planar silicon substrate without nanowire arrays? If the same metal amalgamation immobilization could be formed, why the yield of heterogeneous catalyst based on non-etched flat silicon wafer obviously decreased in cycle 2 and 4 according to the results in ref. 33? If metal amalgamation immobilization could not be formed by using a planar silicon substrate, please give the reason.

The article describes the use of a nanocomposite made by Pd nanoparticles anchored on an array of silicon nanowires in hydrogenation reactions of alkenes. This nanohybrid material shows a remarkable efficiency and proved to be perpetually reusable, as demonstrated by the countless catalytic cycles, due to the robust immobilization of Pd nanoparticle on the silicon support by virtue the formation of Pd-silicides at the interface between the metallic nanoparticle and the silicon wire. Nanohybrid material has been carefully studied by means of common techniques used in these cases (XPS, TEM, SEM, IR and UV-vis). XPS analyses were conducted by sputtering the Pd nanop. surface that revealed the deceasing of peak intensity of Pd(0) with the sputtering time with the simultaneous of the intensity in the higher binding energy region. Transmission IR spectra evidenced the SiNS-Pd and SiNS-H bands, while UV VIS absorption showed no significant change before and after reaction suggesting that the reduction of Pd(II) to Pd(0) did not occur in solution phase but at the interface between Pd(II) solution and the hydrogen terminated silicon surface.

Procedure for hydrogenation is very simple as well as catalyst recovery, and reaction occurs under mild conditions.

Comment. This is a very interesting work that clearly expands results of a previous one by the same Authors (Angew. Chem. Int. Ed. 2014, 53, 127 –131) where the identical nanohybrid material had been used in Heck-Mirozoky couplings, C-H activation reactions and preliminary hydrogenation experiments of some model substrates such as stilbene, nitrobenzene and an unsaturated aldehyde. In this paper, the catalytic material has been more carefully studied and its application extended to the hydrogenation of a wide array of substrates demonstrating that the easy recovery of catalyst is due to the formation of a metallicly-gradated silicon nanocomposite.

This paper introduces significative improvements in hydrogenations procedure that can be of interest to others in the community and the wider field. Results and conclusion are convincing. In my opinion, the paper deserves to be published on this Journal after some minor revision/request listed below:

1. The formation of metallic silicides that renders more robust the catalyst is not a novelty, other groups have highlighted this phenomenon for similar nanohybrids anchoring other metals (see for example Nanomaterials 2018, 8, 78). I suggest integrating references with this one and other similar citations;
2. Experimental procedure shows the absence of stirring during hydrogenation? Is this fact mandatory? Is this due to the possibility of mechanical fracturing by impact of stirring bar with silicon wires on surface? Could a careful stirring increases reaction rate thus reducing reaction time?
3. Preparation of substrate 1b ((Z)-But-2-ene-2,3-diyl)dibenzene) in supplementary material is accomplished in a stereoselective manner with Zn dust an TiCl_4 . I think that the literature should be cited at least for controlling the Z-stereochemistry of olefin compound. The same request for checking the stereochemistry of the reduction product 2b.

Reviewers' comments:

Reviewer #1 (Remarks to the Author):

This paper introduces significant improvements in hydrogenations procedure that can be of interest to others in the community and the wider field. Results and conclusion are convincing. In my opinion, the paper deserves to be published on this Journal after some minor revision/request listed into the attached review file.

The article describes the use of a nanocomposite made by Pd nanoparticles anchored on an array of silicon nanowires in hydrogenation reactions of alkenes. This nanohybrid material shows a remarkable efficiency and proved to be perpetually reusable, as demonstrated by the countless catalytic cycles, due to the robust immobilization of Pd nanoparticle on the silicon support by virtue of the formation of Pd-silicides at the interface between the metallic nanoparticle and the silicon wire. Nanohybrid material has been carefully studied by means of common techniques used in these cases (XPS, TEM, SEM, IR and UV-vis). XPS analyses were conducted by sputtering the Pd nanoparticle surface that revealed the decreasing of peak intensity of Pd(0) with the sputtering time with the simultaneous increase of the intensity in the higher binding energy region. Transmission IR spectra evidenced the SiNS-Pd and SiNS-H bands, while UV VIS absorption showed no significant change before and after reaction suggesting that the reduction of Pd(II) to Pd(0) did not occur in solution phase but at the interface between Pd(II) solution and the hydrogen terminated silicon surface. Procedure for hydrogenation is very simple as well as catalyst recovery, and reaction occurs under mild conditions.

Comment. This is a very interesting work that clearly expands results of a previous one by the same Authors (*Angew. Chem. Int. Ed.* 2014, 53, 127–131) where the identical nanohybrid material had been used in Heck-Mizoroki couplings, C-H activation reactions and preliminary hydrogenation experiments of some model substrates such as stilbene, nitrobenzene and an unsaturated aldehyde. In this paper, the catalytic material has been more carefully studied and its application extended to the hydrogenation of a wide array of substrates demonstrating that the easy recovery of catalyst is due to the formation of a metallicly-graded silicon nanocomposite.

This paper introduces significant improvements in hydrogenations procedure that can be of interest to others in the community and the wider field. Results and conclusion are convincing. In my opinion, the paper deserves to be published on this Journal after some minor revision/request listed below:

1. The formation of metallic silicides that renders more robust the catalyst is not a novelty, other groups have highlighted this phenomenon for similar nanohybrids anchoring other metals (see for

example *Nanomaterials* 2018, 8, 78). I suggest integrating references with this one and other similar citations;

Our Answer: We are thankful to the reviewer's comments. In accordance with the comment, references are added to ref. 35 and 36.

2. Experimental procedure shows the absence of stirring during hydrogenation? Is this fact mandatory? Is this due to the possibility of mechanical fracturing by impact of stirring bar with silicon wires on surface? Could a careful stirring increases reaction rate thus reducing reaction time?

Our Answer: Since silicon nanowire array is readily crashed by mixing with a stirring bar. We used a vortex mixer with temperature controller instead of string bar during reaction. I add information of vortex mixing in the catalytic reactions in Method and supporting information section as follows.

'... under the vortex mixing conditions by a vortex mixer with a temperature controller.'

3. Preparation of substrate 1b ((Z)-But-2-ene-2,3-diyl)dibenzene) in supplementary material is accomplished in a stereoselective manner with Zn dust and TiCl₄. I think that the literature should be cited at least for controlling the Z-stereochemistry of olefin compound. The same request for checking the stereochemistry of the reduction product 2b.

Our Answer: The reaction of acetophenone with Zn dust and TiCl₄ gave a mixture of E/Z-isomers as a crude product. It was difficult to separate the isomers by column chromatography, so that we performed the recrystallization from EtOH to give a single isomer in a moderate yield. We confirmed the isolated compound 1b as the Z-product using X-ray crystallographic analysis, because we were not able to find any reliable publication describing the regiochemistry of the compound. The same is true for the stereochemistry of hydrogenated product 2b. These X-ray crystallographic structures are shown in S7-S8 (supporting information)

Reviewer #2 (Remarks to the Author):

Yamada and coworkers report a methodology of immobilization of a palladium catalyst on to a silicon nanowire

array and its usage in the alkene hydrogenation including tetrasubstituted substrates. The immobilized Pd catalyst is highly active for the hydrogenation of alkenes giving the corresponding saturated products in high yields. They also show an application of hydrogenation of unsaturated fatty acids to form the saturated ones without the contamination by trans-fatty acids. The authors also show the reusability of the catalyst by recycling the catalyst up to 150 times and almost no leaching of the Pd metal by ICP-MS analysis. They propose the formation of Pd-Si alloy (palladium silicide) on the immobilization of the catalyst based on the XPS analysis and XANES spectra of Pd on silicon nanowire array which has low density of Pd on SiNA as a model. Although the authors have previously reported the immobilization method of a metal catalyst on SiNA and its reusability, this article includes some new findings which are worth publishing in Communications Chemistry. Concerning the terms “amalgam” and “amalgamation”, these terms are obscure because these are often used only for the formation of mercury alloy.

Our Answer: I am very glad to receive very positive comments to our research achievement. In accordance with the reviewer's comment, all the word of 'amalgamation' was changed to 'agglomeration' that is a more widely used word.

Reviewer #3 (Remarks to the Author):

This paper reported a perpetually reusable heterogeneous catalyst for alkene hydrogenation through metal amalgamation immobilization of Pd nanoparticles on silicon nanowire arrays. Though the heterogeneous catalyst of SiNA-Pd have been used for catalytic organic transformations in the author's previous works (ref. 33 in this paper), the author found this heterogeneous catalyst could be reused without yield-loss for more than one hundred times and investigated the mechanism for the stability in this paper. This metal amalgamation immobilization may be applied to the preparation of a variety of metal nanoparticle catalysts with high stability. The results are interesting. I think this paper should be published after addressing the following comments:

1. The scheme of fabricating processes as shown in fig.2a is the same as scheme 1 in ref. 33.

Our Answer: We are thankful to the reviewer's comments. Since this procedure in Figure 2, a newly prepared for this manuscript on the basis of Scheme 1 in ref.33 as the reviewer kindly pointed out, is important to discuss how palladium nanoparticles are deposited in silicon nanowires, we put this figure here in Figure 2.

2. The characterization of the heterogeneous catalyst of SiNA-Pd is inadequate. Both magnified SEM and TEM images are needed to intuitively show the structures of the heterogeneous catalyst. Especially, high-resolution

TEM may show the lattice transition from silicon to Pd particles.

Our Answer: Thank you very much for the reviewer's comment. In accordance with the reviewer's comment, we added SEM & TEM images to supporting information in Pages 11 and 12.

3. The authors showed compared results of Pd/C catalysts in Fig. 3, but didn't show the structure of the Pd/C catalyst or the related description of the contrast experiment. In addition, why does the yield of Pd/C catalyst sharply decrease after 10 cycles?

Our Answer: We checked ICP-MS to check leaching of Pd in the reaction mixture. When the Pd/C (5wt%), purchased from Fujifilm Wako Chemicals, was reused 5 times, still 233 ppm of Pd was leached out to the reaction mixture. I suppose this is why the catalytic activity of Pd/C decreased during its reusing.

4. Could the heterogeneous catalyst be formed through metal amalgamation immobilization by using a planar silicon substrate without nanowire arrays? If the same metal amalgamation immobilization could be formed, why the yield of heterogeneous catalyst based on non-etched flat silicon wafer obviously decreased in cycle 2 and 4 according to the results in ref. 33? If metal amalgamation immobilization could not be formed by using a planar silicon substrate, please give the reason.

Our Answer: We have no evidence whether the same metal amalgamation immobilization occurs or not. From the results in ref. 33, I suppose that at least suitable deposition of Pd should not occur.

Our results suggest that silicon nanowire array control the formation of Pd/Si nanoparticles with suitable size. Since flat silicon wafer has no control nanospace, too large metal amalgamation or no amalgamation occurs in the flat silicon wafer. Therefore, in this manuscript, we used SiNA-Pd for the catalytic reactions.

REVIEWERS' COMMENTS:

Reviewer #1 (Remarks to the Author):

Authors answered satisfyingly to the addressed points. The paper can be published in the present form.

Reviewer #3 (Remarks to the Author):

The authors have well addressed the comments about the referees and I think this paper can be accepted in this version from my point.